# Learning Likelihoods with Conditional Normalizing Flows

## Abstract

Normalizing Flows (NFs) are able to model complicated distributions $p_Y(y)$ with strong inter-dimensional correlations and high multimodality by transforming a simple base density $p_Z(z)$ through an invertible neural network under the change of variables formula. Such behavior is desirable in multivariate structured prediction tasks, where handcrafted per-pixel loss-based methods inadequately capture strong correlations between output dimensions. We present a study of conditional normalizing flows (CNFs), a class of NFs where the base density to output space mapping is conditioned on an input x, to model conditional densities $p_{Y|X}(y|x)$. CNFs are efficient in sampling and inference, they can be trained with a likelihood-based objective, and CNFs, being generative flows, do not suffer from mode collapse or training instabilities. We provide an effective method to train continuous CNFs for binary problems and in particular, we apply these CNFs to super-resolution and vessel segmentation tasks demonstrating competitive performance on standard benchmark datasets in terms of likelihood and conventional metrics.

## 1 Introduction

Learning conditional distributions $p_{Y|X}(\mathbf{y}|\mathbf{x})$ is one of the oldest problems in machine learning. When the output $\mathbf{y}$ is high-dimensional this is a particularly challenging task, and the practitioner is left with many design choices. Do we factorize the conditional? If not, do we model correlations with, say, a conditional random field (Prince, 2012)? Do we use a unimodal distribution? How fat should the tails be? Do we use an explicit likelihood at all, or use implicit methods (Mohamed & Rezende, 2015) such as a GAN (Goodfellow et al., 2014)? Do we quantize the output? Ideally, the practitioner should not have to make design choices at all, and the distribution should be *learned* from the data.

In the field of density estimation *normalizing flows* (NFs) are a relatively new family of models (Rezende & Mohamed, 2015). NFs model complicated high dimensional *marginal* distributions $p_Y(\mathbf{y})$ by transforming a simple *base distribution* or *prior* $p_Z(\mathbf{z})$ through a learnable, invertible mapping $f_\phi$ and then applying the change of variables formula. NFs are efficient in inference and sampling, are able to learn inter-dimensional correlations and multi-modality, and they are exact likelihood models, amenable to gradient-based optimization.

Flow-based generative models (Dinh et al., 2016) are generally trained on the image space, and are in some cases computationally efficient in both the forward *and* inverse direction. These are advantageous over other likelihood based methods because *i)* sampling is efficient opposed to autoregressive models (Van Oord et al., 2016), and *ii)* flows admit exact likelihood optimization in contrast with variational autoencoders (Kingma & Welling, 2014).

Conditional random fields directly model correlations between pixels, and have been fused with deep learning (Chen et al., 2016). However, they require the practitioner to choose which pixels have pairwise interactions. Another approach uses adversarial training (Goodfellow et al., 2014). A downside is that the training procedure can be unstable and they are difficult to evaluate quantitatively.

We propose to learn the likelihood of conditional distributions with few modeling choices using *Conditional Normalizing Flows* (CNFs). CNFs can be harnessed for conditional distributions $p_{Y|X}(\mathbf{y}|\mathbf{x})$

by conditioning the prior and the invertible mapping on the input $\mathbf{x}$. In particular, we apply conditional flows to super-resolution (Wang et al., 2018) and vessel segmentation (Staal et al., 2004). We evaluate their performance gains on multivariate prediction tasks along side architecturally-matched factored baselines by comparing *likelihood* and application specific evaluation metrics.

## 2 BACKGROUND

In the following, we present the relevant background material on normalizing flows and structured prediction. This section covers the change of variables formula, invertible modules, variational dequantization and conventional likelihood optimization.

### 2.1 NORMALIZING FLOWS

A standard NF in continuous space is based on a simple change of variables formula. Given two spaces of equal dimension $\mathcal{Z}$ and $\mathcal{Y}$; a once-differentiable, parametric, bijective[1] mapping $f_\phi : \mathcal{Y} \to \mathcal{Z}$, where $\phi$ are the *parameters* of $f$; and a *prior* distribution $p_Z(\mathbf{z})$, we can model a *complicated distribution* $p_Y(\mathbf{y})$ as

$$p_Y(\mathbf{y}) = p_Z(f_\phi(\mathbf{y})) \left| \frac{\partial f_\phi(\mathbf{y})}{\partial \mathbf{y}} \right|. \tag{1}$$

The term $|\partial f_\phi(\mathbf{y})/\partial \mathbf{y}|$ is the *Jacobian determinant* of $f_\phi$, evaluated at $\mathbf{y}$ and it accounts for volume changes induced by $f_\phi$. The transformation $f_\phi$ introduces correlations and multi-modality in $p_Y$. The main challenge in the field of normalizing flows is designing the transformation $f_\phi$. It has to be *i)* bijective, *ii)* have an efficient and tractable Jacobian determinant, *iii)* be from a 'flexible' model class. In addition, *iv)* for fast sampling the inverse needs to be efficiently computable. Below we briefly state which invertible modules are used in our architectures, obeying the aforementioned points.

**Coupling layers**  Affine coupling layers (Dinh et al., 2016) are invertible, nonlinear layers. They work by splitting the input $\mathbf{z}$ into two components $\mathbf{z}_0$ and $\mathbf{z}_1$ and nonlinearly transforming $\mathbf{z}_0$ as a function of $\mathbf{z}_1$, before reconcatenating the result. If $\mathbf{z} = [\mathbf{z}_0, \mathbf{z}_1]$ and $\mathbf{y} = [\mathbf{y}_0, \mathbf{y}_1]$ this is

$$\mathbf{y}_0 = s(\mathbf{z}_1) \cdot \mathbf{z}_0 + t(\mathbf{z}_1) \qquad\qquad \mathbf{z}_0 = (\mathbf{z}_0 - t(\mathbf{y}_1))/s(\mathbf{y}_1)$$
$$\mathbf{y}_1 = \mathbf{z}_1 \qquad\qquad \mathbf{z}_1 = \mathbf{y}_1$$

Where the scale $s(\cdot)$ and translation $t(\cdot)$ functions can be any function, typically implemented with a CNN. Similar conditioning with normalizing flows has been done in previous works by Mohamed & Rezende (2015) and Kingma et al. (2016b).

**Invertible 1 x 1 Convolutions**  Proposed in Kingma & Dhariwal (2018), invertible $1 \times 1$ convolutions help mix information across channel dimensions. We implement them as regular $1 \times 1$ convolutions and for the inverse, we convolve with the inverse of the kernel.

**Squeeze layers**  Squeeze layers (Dinh et al., 2016) are used to compress the spatial resolution of activations. These also help with increasing spatial receptive field of pixels in the deeper activations.

**Split Prior**  Split priors (Dinh et al., 2016) work by splitting a set of activations $\mathbf{z}$ into two components $\mathbf{z}_0$ and $\mathbf{z}_1$. We then condition $\mathbf{z}_1$ on $\mathbf{z}_0$ using a simple base density e.g. $p(\mathbf{z}_1|\mathbf{z}_0) = \mathcal{N}(\mathbf{z}_1; \mu(\mathbf{z}_0), \sigma^2(\mathbf{z}_0))$, where $\mu(\cdot)$ and $\sigma^2(\cdot)$ are neural networks. The component $\mathbf{z}_0$ can be modeled by further flow layers. This prior, is useful for modeling hierarchical correlations between dimensions, and also helps reduce computation, since $\mathbf{z}_0$ is reduced in size.

**Variational dequantization**  When modeling discrete data, Theis et al. (2016) introduced the concept of *dequantization*. For this, they modeled the probability mass function over $\mathbf{y}$ as a latent variable model

$$P_{\text{model}}(\mathbf{y}) = \int_{\mathcal{V}} P(\mathbf{y}|\mathbf{v})p(\mathbf{v}) \, \mathrm{d}\mathbf{v} = \int_{\mathcal{V}} p(\mathbf{y}, \mathbf{v}) \, \mathrm{d}\mathbf{v} \tag{2}$$

---

[1]We can also use injective mappings

where the latent variables $\mathbf{v} \in \mathcal{V}$ are continuous-valued. This is a convenient model to use, since the marginal $p(\mathbf{v})$, living on a continuous sample space, can be modelled with a continuous NF. The distribution $P(\mathbf{y}|\mathbf{v})$ is known as the *quantizer* and is typically an indicator function $P(\mathbf{y}|\mathbf{v}) = \mathbb{I}[\mathbf{v} \in \mathbf{y} + [0,1)^D]$. Other works (Hoogeboom et al., 2019a; Tran et al., 2019) directly model $P_{\text{model}}(\mathbf{y})$ with a discrete-valued flow, but these are known to be difficult to optimize. As an extension of dequantization, Ho et al. (2019) introduced a variational distribution $q(\mathbf{v}|\mathbf{y})$, called a *dequantizer*, and write a lower bound on the data log-likelihood using Jensen's inequality as follows

$$\mathbb{E}_{P_{\text{data}}(\mathbf{y})} \log P_{\text{model}}(\mathbf{y}) = \mathbb{E}_{P_{\text{data}}(\mathbf{y})} \log \int p(\mathbf{y}, \mathbf{v}) \, \mathrm{d}\mathbf{v} \tag{3}$$

$$= \mathbb{E}_{P_{\text{data}}(\mathbf{y})} \log \int \frac{q(\mathbf{v}|\mathbf{y})}{q(\mathbf{v}|\mathbf{y})} p(\mathbf{y}, \mathbf{v}) \, \mathrm{d}\mathbf{v} \geq \mathbb{E}_{P_{\text{data}}(\mathbf{y})} \int q(\mathbf{v}|\mathbf{y}) \log \frac{p(\mathbf{y}, \mathbf{v})}{q(\mathbf{v}|\mathbf{y})} \, \mathrm{d}\mathbf{v} \tag{4}$$

Noting that the joint $p(\mathbf{y}, \mathbf{v}) = \mathbb{I}[\mathbf{v} \in \mathbf{y} + [0,1)^D]p(\mathbf{v})$, we see that the dequantizer distribution $q(\mathbf{v}|\mathbf{y})$ must be defined such that $\mathbf{v} \in \mathbf{y} + [0,1)^D$, otherwise $p(\mathbf{y}, \mathbf{v}) = 0$ and the lower-bound is undefined. Restricting $q(\mathbf{v}|\mathbf{y})$ to satisfy this condition, results in the following variational dequantization bound

$$\mathbb{E}_{P_{\text{data}}(\mathbf{y})} \log P_{\text{model}}(\mathbf{y}) \geq \mathbb{E}_{P_{\text{data}}(\mathbf{y})} \int q(\mathbf{v}|\mathbf{y}) \log \frac{p(\mathbf{v})}{q(\mathbf{v}|\mathbf{y})} \, \mathrm{d}\mathbf{v}. \tag{5}$$

## 2.2 STRUCTURED PREDICTION

Structured prediction tasks such as image segmentation or super-resolution, can be probabilistically framed as learning an unknown target distribution $p^*(\mathbf{y}|\mathbf{x})$, with an input $\mathbf{x} \in \mathcal{X}$ and a target $\mathbf{y} \in \mathcal{Y}$. In practice with deep learning models, the unknown distribution is often learned by a factored model:

$$p(\mathbf{y}|\mathbf{x}) = \prod_{d=1}^{D} p(y_d|\mathbf{x}), \tag{6}$$

where $y_d$ represents the $d$th dimension of $\mathbf{y}$. Several loss-based optimization methods are a special case of this factored model. The mean squared error is equivalent to a product of normal distributions with equal and fixed standard deviation. Other examples are: cross entropy, equivalent to a product of log categorical distributions, and binary cross entropy, equivalent to a product of log Bernoulli distributions.

With factorized *independent* likelihoods, individual dimensions of $\mathbf{y}$ are assumed to be conditionally independent. As a result, sampling leads to results with uncorrelated noise over the output dimensions. In the literature, a fix for this problem is to visualize the mode of the distribution and interpret that as a prediction. However, because the likelihood was optimized assuming a conditionally independent noise distribution, these modes tend to be blurry and lack crisp details.

## 3 METHOD

In this section we present our main innovations. *i)* learning conditional likelihoods using CNFs and *ii)* a variational dequantization framework for binary random variables.

## 3.1 CONDITIONAL NORMALIZING FLOWS

We propose to learn conditional likelihoods using conditional normalizing flows for complicated target distributions in multivariate prediction tasks. Take an input $\mathbf{x} \in \mathcal{X}$ and a regression target $\mathbf{y} \in \mathcal{Y}$. We learn a complicated distribution $p_{Y|X}(\mathbf{y}|\mathbf{x})$ using a conditional prior $p_{Z|X}(\mathbf{z}|\mathbf{x})$ and a mapping $f_\phi : \mathcal{Y} \times \mathcal{X} \rightarrow \mathcal{Z}$, which is bijective in $\mathcal{Y}$ and $\mathcal{Z}$. The likelihood of this model is:

$$p_{Y|X}(\mathbf{y}|\mathbf{x}) = p_{Z|X}(\mathbf{z}|\mathbf{x}) \left| \frac{\partial \mathbf{z}}{\partial \mathbf{y}} \right| = p_{Z|X}(f_\phi(\mathbf{y}, \mathbf{x})|\mathbf{x}) \left| \frac{\partial f_\phi(\mathbf{y}, \mathbf{x})}{\partial \mathbf{y}} \right|. \tag{7}$$

Notice that the difference between Equations 1 and 7 is that all distributions are conditional and the flow has a conditioning argument of $\mathbf{x}$.

The generative process from $\mathbf{x}$ to $\mathbf{y}$ (shown in Figure 1) can be described by first sampling $\mathbf{z} \sim p_{Z|X}(\mathbf{z}|\mathbf{x})$ from a simple base density with its parameters conditioned on $\mathbf{x}$ (for us this is a diagonal Gaussian) and then passing it through a sequence of bijective mappings $f_\phi^{-1}(\mathbf{z}; \mathbf{x})$. This allows for modelling multimodal conditional distributions in $\mathbf{y}$, which is typically uncommon.

For the training procedure, the process runs in reverse. We begin with label $\mathbf{y}$ and conditioning input $\mathbf{x}$. We 'flow' the label back through $f_\phi$ to yield $\mathbf{z} = f_\phi(\mathbf{y}; \mathbf{x})$, and then we evaluate the log-likelihood of the parameters of the prior, given this transformed label $\mathbf{z}$. The flow and prior parameters can be optimized using stochastic gradient descent and training in minibatches in the usual fashion. Note that this style of training a conditional density model $p_{Y|X}(\mathbf{y}|\mathbf{x})$ differs fundamentally from traditional models, because we compute the log-likelihood in $\mathbf{z}$-space and not $\mathbf{y}$-space. As a result, we are not biasing our results with an arbitrary choice of output-space likelihood or in the case of this paper, handcrafted image loss. Instead, one could interpret this method as learning the correlational and multimodal structure of the likelihood or simply put loss-learning.

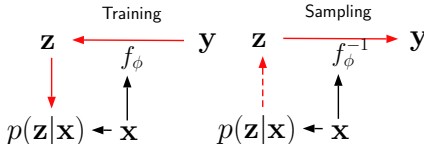

Figure 1: *Diagram of our model in the train and sampling phases. Solid lines represent deterministic mappings and dashed lines represent sampling. The conditioning variable enters the network in base density $p(\mathbf{z}|\mathbf{x})$ and the bijective mappings $f(\mathbf{y}, \mathbf{x})$.*

**Conditional modules** In our work, the conditioning is introduced in the prior, the split priors, and the affine coupling modules. For the prior, we set the mean and variance as functions of $\mathbf{x}$. For the split prior, we add $\mathbf{x}$ as a conditoning argument to the conditional $p(\mathbf{z}_1|\mathbf{z}_0, \mathbf{x})$. And for the affine coupling layers, we pass $\mathbf{x}$ to the scale and translation networks so that

| | |
|---|---|
| Conditional Prior | $p(\mathbf{z}|\mathbf{x}) = \mathcal{N}(\mathbf{z}; \mu(\mathbf{x}), \sigma^2(\mathbf{x}))$ |
| Conditional Split Prior | $p(\mathbf{z}_1|\mathbf{z}_0, \mathbf{x}) = \mathcal{N}(\mathbf{z}_1; \mu(\mathbf{z}_0, \mathbf{x}), \sigma^2(\mathbf{z}_0, \mathbf{x}))$ |
| Conditional Coupling | $\mathbf{y}_0 = s(\mathbf{z}_1, \mathbf{x}) \cdot \mathbf{z}_0 + t(\mathbf{z}_1, \mathbf{x}); \qquad \mathbf{y}_1 = \mathbf{z}_1$ |

In practice these functions are implemented using deep neural networks. First the conditioning term $\mathbf{x}$ is transformed into a rich representation $\mathbf{h} = g(\mathbf{x})$ using a large network $g$. Subsequently, each function in the flow is applied to a concatenation $[\cdot, \cdot]$ of $\mathbf{h}$ and the relevant part of $\mathbf{z}$. For example, the translation of a conditional coupling is computed as $t(\mathbf{z}_1, \mathbf{x}) = \text{NN}([\mathbf{z}_1, \mathbf{h}])$.

### 3.2 Variational Dequantization for Binary Random Variables

We generalize the variational dequantization scheme for the binary setting. Let $\mathbf{y} \in \{0, 1\}^D$ be a multivariate binary random variable and $\mathbf{v} \in \mathbb{R}^D$ its *dequantized* representation. In Ho et al. (2019) the bound is not guaranteed to be tight, since there is a domain mismatch in the support of $p(\mathbf{v})$ and the variational dequantizer $q(\mathbf{v}|\mathbf{y})$. Technically, if $p(\mathbf{v})$ is modeled as a bijective mapping from a Gaussian distribution where the mapping only has finite volume changes, then the support of $p(\mathbf{v})$ is unbounded. On the otherhand, the support of the dequantizer is bounded and so we have to redefine either the dequantizer to map to all of $\mathbb{R}^D$ or restrict the support of the flow to a bounded volume inside $\mathbb{R}^D$. We resolve this by dequantizing with half-infinite noise, where

$$\mathbf{v}|\mathbf{y}, \mathbf{z} = 0.5 + \text{sign}(\mathbf{y} - 0.5) \cdot \text{softplus}(\text{NN}(\mathbf{z})). \tag{8}$$

The softplus guarantees that samples from the neural network NN are only positive. If $\mathbf{y}$ is 1, the term $\text{sign}(\mathbf{y} - 0.5)$ outputs positive-valued noise and if $\mathbf{y}$ is 0 the noise is negative-valued.

## 4 Related Work

Normalizing flows were originally introduced to machine learning to learn a flexible variational posterior, a conditional distribution, in VAEs (Rezende & Mohamed, 2015; Kingma et al., 2016a; van den Berg et al., 2018). Flow-based generative models (Dinh et al., 2016; Papamakarios et al., 2017; Huang et al., 2018; Kingma & Dhariwal, 2018; Hoogeboom et al., 2019b; Grathwohl et al., 2019; Cao et al., 2019; Chen et al., 2019) are typically trained directly in the data space. Several of these are designed to be fast to invert, which makes them suitable for drawing samples after training.

Different versions and applications of conditional normalizing flows include Agrawal & Dukkipati (2016) who utilize flows in the decoder of Variational AutoEncoders Kingma & Welling (2014), which are conditioned on the latent variable. Trippe & Turner (2018) who utilize conditional flows for prediction problems in a Bayesian framework for density estimation. Atanov et al. (2019) introduce a semi-conditional flow that provides an efficient way to learn from unlabeled data for semi-supervised classification problems. Very recently, Ardizzone et al. (2019) have proposed conditional flow-based generative models for image colorization, which differs from our work in training objective, architecture and applicability to binary segmentation. Autoregressive models (Van Oord et al., 2016) have also been studied for conditional image generation van den Oord et al. (2016) but are generally slow to sample from.

Adversarial methods (Goodfellow et al., 2014) have widely been applied to (conditional) image density modeling tasks (Vu et al., 2019; Sajjadi et al., 2017b; Yuan et al., 2018; Mechrez et al., 2018), because they tend to generate high-fidelity images. Disadvantages of adversarial methods are that they can be complicated to train, and it is difficult to obtain likelihoods. For this reason, it can be hard to assess whether they are overfitting or generalizing.

## 5 EXPERIMENTS

Here we explain our experiments into super-resolution and vessel segmentation. All models were implemented using the PyTorch framework.

### 5.1 SINGLE IMAGE SUPER RESOLUTION

Single Image Super Resolution (SISR) methods aims to find a high resolution image $x_{hr}$ given a single (downsampled) low resolution image $x_{lr}$. Framing this problem as learning a likelihood, we utilize a CNF to learn the distribution $p(x_{hr}|x_{lr})$. To compare our method we also train a factorized baseline likelihood model with comparable architectures and parameter budget. The factorized baseline uses a product of discretized logistic distributions (Kingma et al., 2016a; Salimans et al., 2017). All methods are compared on negative $\log_2$-likelihood if available, which has the information theoretic interpretation *bits per dimensions*. In addition, we evaluate using SSIM (Wang et al., 2004) and PSNR metrics.

**Implementation Details**  The flow is based on the (Dinh et al., 2016; Kingma & Dhariwal, 2018) multi-scale architectures. Each step of flow consists of $K$ subflows and $L$ levels. One subflow consists of an activation normalization, $1 \times 1$ convolution, and our conditional coupling layer. After completing a level, half of the representation is factored-out and modeled using our conditional split prior. After all levels have been completed, our conditional prior is used to model the final part of the latent variable.

The conditioning variable $x_{lr}$ is transformed into the feature representation $\mathbf{h}$ using Residual-in-Residual Dense Block (RRDB) architecture (Wang et al., 2018), consisting of 16 residual-in-residual blocks. To match the parameter budget, the channel growth is 55 for the baseline and the growth is 32 for the CNF.

**Data**  The models are trained on natural image datasets, Imagenet32 and Imagenet64 (Chrabaszcz et al., 2017). Since the dataset has no test set, we use its validation images as a test set. For validation we take 10000 images from the train images. The performance is always reported on the test set unless specified otherwise. We evaluate our models on widely used benchmark datasets Set5 (Bevilacqua et al., 2012), Set14 (Zeyde et al., 2012) and BSD100 (Huang et al., 2015). At test time, we pad the test images with zeros at right and bottom so that they are square and compatible with squeeze layers. When evaluating on SSIM and PSNR, we can extract the patch with the exact image shape. For all datasets the LR images are obtained using MATLABs bicubic kernel function with reducing aliasing artifacts following Wang et al. (2018). For these experiments, the pixel values are dequantized by adding uniform noise (Theis et al., 2016).

**Training Settings**  We train on ImageNet32 and ImageNet64 for $200,000$ iterations with mini batches of size 64, and a learning rate of 0.0001 using the Adam optimizer (Kingma & Ba, 2015).

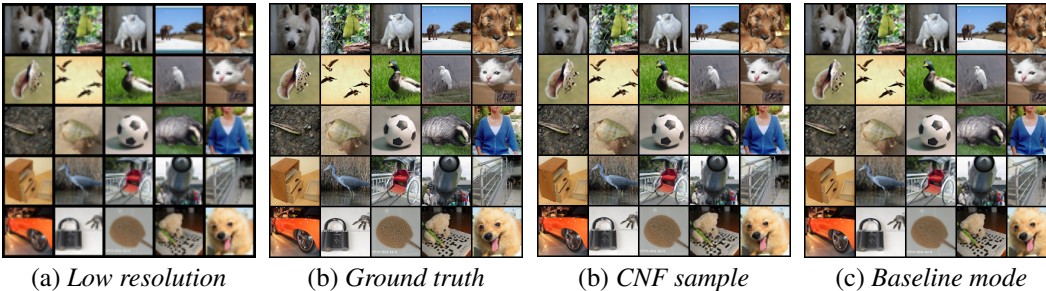

| (a) *Low resolution* | (b) *Ground truth* | (b) *CNF sample* | (c) *Baseline mode* |

Figure 2: Super resolution results on the Imagenet64 test data. Samples are taken from the CNF $x_{hr} \sim p(x_{hr}|x_{lr})$ and the mode is visualized for the factorized baseline model. *Best viewed electronically.*

The high-resolution image $x_{hr}$ either the original $32 \times 32$ or $64 \times 64$ original input images. The flow architecture is build with $L = 2$ levels and $K = 8$.

### 5.1.1 EVALUATION

In this section the performance of CNFs for SISR is compared against a baseline likelihood model on ImageNet32 and ImageNet64. Their performance measured in $\log_2$-likelihood (bits per dimension) is shown in Table 1, which show that the CNF outperforms the factorized baseline in likelihood. Recall that the baseline model is factorized and conditionally independent. These results indicate that it is advantageous to capture the correlations and multi-modality present in the data.

Table 1: Comparison of likelihood learning with CNFs and factorized discrete baseline on ImageNet32 and ImageNet64 measured in bits per dimensions.

| Dataset | CNF | factorized LL |
|---|---|---|
| ImageNet32 | **3.01** | 4.00 |
| ImageNet64 | **2.90** | 3.61 |

Super resolution samples $x_{hr} \sim p(x_{hr}|x_{lr})$ from the Imagenet64 test data are shown in Figure 2. The distribution mode of the factorized baseline is displayed in Figure 2 panel c). We show that the baseline is able to learn a relationship between conditioning variable **x** and **y**, but lacks crisp details. The super-resolution images from the CNF are shown in panel b). Notice there are more high-frequency components modelled, for instance in grass and in hairs. Following Kingma & Dhariwal (2018), we sample from the base distributions with a temperature $\tau$ of 0.8 to achieve the best perceptual quality for the distribution learned by the CNF.

As there are no standard metric for measuring perceptual quality, we measure performance results on PSNR and SSIM between our predicted image and the ground truth image in Table 2. We compare CNFs to other state-of-the-art per-pixel loss based methods and the factorized baseline for a 2x upsampling task on standard super-resolution benchmarks. If available, we report negative $\log_2$-likelihood (bpd) (computed as an average over 1000 randomly cropped 128 x 128 patches). Note that without any hyperparameter tuning or compositional loss weighting, as is typical in SISR, the CNF performs competitively with state-of-the-art super-resolution methods by simply optimizing the likelihood. The SSIM scores for the baseline perform on par or better than the adversarial methods and the CNF on all benchmarks. On PSNR scores however, the CNF beats the factorized discrete baseline. Samples shown in Figure 3 show that the CNF predictions have more fine grained texture details. Comparing this finding with the baseline that outperforms every method on SSIM, show that metrics can be misleading.

Notice how samples from a independent factorized likelihood model have a lot of color noise, whereas samples from the CNF do not have such problems. Increasing temperature increases high-level detail, where we find that $\tau = 0.5$ strikes a balance between noise smoothing and detail. This can be attributed to the property of flows to model pixel correlations among output dimensions for high-dimensional data such as images.

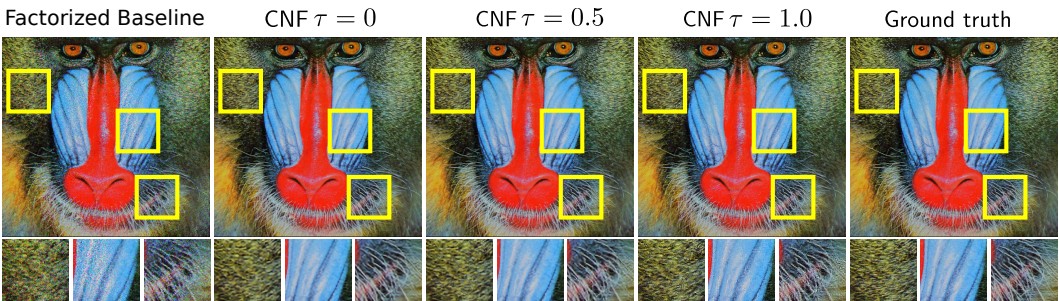

Figure 3: Conditional samples from the CNF (ours) for sampling temperatures $\{0., 0.5, 1.0\}$ and the factorized discrete baseline for 2x upscaling. Conditioning image is a baboon from Set14 test set. Both models were trained on ImageNet64. *Best viewed electronically.*

Table 2: CNF compared to factorized discrete baseline and adversarial, pixel-wise methods (Dong et al., 2015; Sajjadi et al., 2017a; Prez-Pellitero et al., 2016) based on negative $\log_2$-likelihood (bits per dimension or bpd), PSNR, SSIM and for 2x upscaling. Our methods were trained on ImageNet64.

| Model Type | Set5 | | | Set14 | | | BSD100 | | |
|---|---|---|---|---|---|---|---|---|---|
| | bpd | PSNR | SSIM | bpd | PSNR | SSIM | bpd | PSNR | SSIM |
| Bicubic | - | 33.7 | 0.930 | - | 30.2 | 0.869 | - | 29.6 | 0.843 |
| SRCNN | - | 36.7 | 0.954 | - | 32.4 | 0.906 | - | 31.4 | 0.888 |
| PSyCO | - | 36.9 | 0.956 | - | 32.6 | 0.898 | - | 31.4 | 0.890 |
| ENet | - | **37.3** | **0.958** | - | **33.3** | 0.915 | - | **32.0** | 0.898 |
| LL Baseline | 2.34 | 32.5 | **0.958** | 3.23 | 31.0 | **0.917** | 3.20 | 30.6 | **0.900** |
| CNF (ours) | **2.11** | 36.2 | 0.957 | **2.51** | 32.5 | 0.911 | **2.33** | 31.4 | 0.893 |

## 5.2 VESSEL SEGMENTATION

Vessel segmentation is an important, long-standing, medical imaging problem, where we seek to segment blood vessels from pictures of the retina (the back of the eye). This is a difficult task, because the vessels are thin and of varying thickness. A likelihood function used in segmentation is

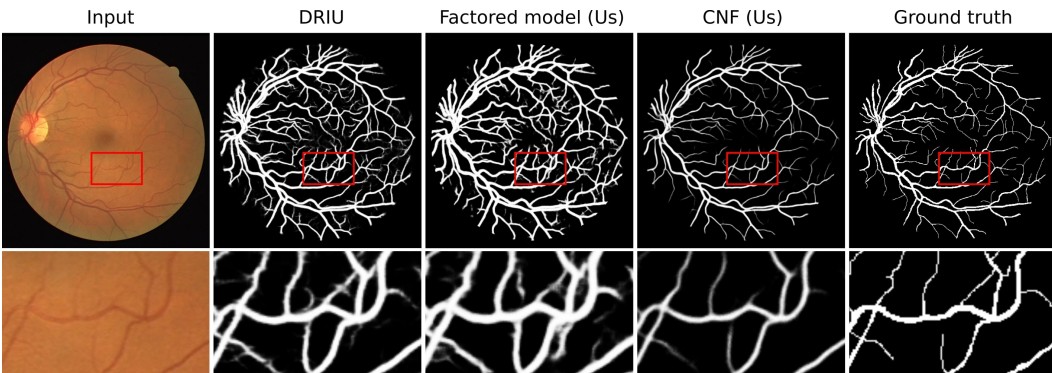

Figure 4: Example of retinal segmentations using DRIU, our likelihood baseline trained with the same loss, and our CNF. For the CNF, the mean of 100 samples is visualized. Notice that our segmentations more accurately capture the vessel width, which is overdilated in the DRIU and factored models.

Table 3: Numerical results on the DRIVE dataset. We see that the CNF is in the range of the SOTA model DRIU. SE: Structured Forests (Dollár & Zitnick, 2013), LD: Line Detector (Ricci & Perfetti, 2007), Wavelets (Soares et al., 2006), Human (Staal et al., 2004), HED: Holistic Edge Detector (Xie & Tu, 2015), KB: Kernel Boost Becker et al. (2013), $N^4$: $N^4$ Fields (Ganin & Lempitsky, 2014), DRIU: Deep Retinal Image Understanding (Maninis et al., 2016). Our answers are shown in mean $\pm$ 1std form, where statistics are taken over 5 runs.

| | SE | LD | Wavelets | Human | HED | KB | $N^4$ | DRIU | Factored (ours) | CNF Uniform (Ours) | CNF (ours) |
|---|---|---|---|---|---|---|---|---|---|---|---|
| bpd | - | - | - | - | - | - | - | - | $0.0647 \pm 0.0015$ | $0.3366 \pm 0.0290$ | $\mathbf{0.0254} \pm 0.0008$ |
| F-Score | 0.658 | 0.692 | 0.762 | 0.791 | 0.794 | 0.800 | 0.805 | **0.821** | $0.815 \pm 0.001$ | $0.762 \pm 0.002$ | $0.819 \pm 0.001$ |

a weighted Bernoulli distribution

$$\mathcal{L} = \prod_j \frac{p_j^{\beta \cdot y_j}(1 - p_j)^{(1-\beta)\cdot(1-y_j)}}{p_j^\beta + (1 - p_j)^{1-\beta}} \tag{9}$$

where $p_j = p(y_j = 1|\mathbf{x})$ is the prediction probability that pixel $y_j$ is positive (vessel class) and $\beta$ is a class balancing constant set to $\sim 10\%$ for us. This loss function is preferred, because it accounts for the apparent class imbalance in the ratio of vessels to background. In practice, the numerator of this likelihood is used as a loss function and the normalizer is ignored. The resulting loss is called a weighted cross-entropy. In our experiments we train using the weighted cross-entropy (as in the literature), but we report likelihood values including the normalizer, for a meaningful comparison.

**Dataset and comparisons** We test on the DRIVE database Staal et al. (2004) consisting of $584 \times 565$, 8-bit RGB images, split into 20 train, and 20 test images. To compare against other methods, we plot precision-recall curves, report the maximum F-score along each curve (shown as a dot in the graph), report the bits per dimension, and plot distributions in PR-space. The main CNN-based contenders are Deep Retinal Image Understanding (DRIU) (Maninis et al., 2016) and Holistically-Nested Edge Detection (HED) (Xie & Tu, 2015), both of which are instances of Deeply-Supervised Nets (Lee et al., 2015). The main difference between DRIU and HED is that DRIU is pretrained on ImageNet Krizhevsky et al. (2012); whereas, HED is not. Other competing methods are reported with results cited from (Maninis et al., 2016). For fairness, we also train a model which we call the *likelihood baseline*, which uses the exact same architecture as the flow but run a feedforward model and trained with the weighted Bernoulli loss.

**Implementation** The flow is identical to the flow used in the previous section, with some key differences. *i)* instead of activation normalization, we use instance normalization (Ulyanov et al., 2016), *ii)* the conditional affine coupling layers do not contain a scaling component $s(\cdot)$ but just the translation $t(\cdot)$, hence it is volume preserving, and *iii)* we train using variational dequantization. Since the data is binary-valued, we dequantize according to the Flow++ scheme of Ho et al., modified to binary variables (see Section 3.2), using a CNF at just a single scale for the dequantizer. The CNF is conditioned on resolution-matched features extracted from a VGG-like network Simonyan & Zisserman (2015). This model is composed of blocks of the form block = [InstanceNorm, ReLU, conv], and 2x2 max-pooling layer, shown in Table 4. All filter sizes are 3x3. The outputs are at layers 4 and 7. These are used to condition the resolution 512 and 256 levels of the CNF, respectively.

Table 4: Feature extractor architecture for retinal vessel segmentation. RES. abbreviates resolution. Outputs are at layer 4 and 7.

| LAYER | TYPE | RES. |
|---|---|---|
| 0 | input | 1024 |
| 1 | block | 1024 |
| 2 | max-pool | 512 |
| 3 | block | 512 |
| 4 | block | 512 |
| 5 | max-pool | 256 |
| 6 | block | 256 |
| 7 | block | 256 |

**Training/test settings** We train using the Adam optimizer at learning rate 0.001, and a minibatch size of 2 for 2000 epochs. All images are padded to 1024x1024 pixels, so that they are compatible with squeeze layers. We use $360°$ rotation augmentation, isotropic scalings in the range $[0.8, 1.2]$, and shears drawn from a normal distribution with standard deviation $10°$. At test time we draw samples from our model and compare those against the groundtruth labels. This contrasts with other methods, that measure labels against thresholded versions of a factorized predictive distribution. To

create the PR curve in Figure 5 we take the average of 100 samples and threshold the resulting map (example shown in Figure 4). While crude, this mean image is useful in defining a PR-curve, since there is not great topology change between samples.

**Evaluation**    The results of our experiments are shown in Table 3 and Figure 5, with a visualization in Figure 4. We see in the table that the CNF trained with our binary dequantization achieves the best bits per dimension, with comparable F-score to the state of the art model (DRIU), but our model does not require pretraining on ImageNet. Interestingly, we found training a flow with uniform dequantization slightly unstable and the results were far from satisfactory. In the PR-curve Figure 5, we show a comparable curve for our binary dequantized CNF to the DRIU model. These results, however, say nothing about the calibration of the probability outputs, but just that the various probability predictions are well ranked. To gain an insight into the calibration of the probabilities, we measure the distribution of precision and recall values for point samples drawn from all models, including a second human grader, present in the original DRIVE dataset. We synthesized samples from factored distributions (all except ours and 'human'), by sampling images from a factored Bernoulli with mean as the soft image. We see the results in the right hand plot of Figure 5, which shows that while the other CNN-based methods such as DRIU or HED have good precision, they suffer in term of recall. On the other hand, the CNF drops in precision a little bit, but makes up for this in terms of high recall, with a PR distribution overlapping the human grader. This indicates that the CNF has learned a well-calibrated distribution, compared to the baseline methods. Further evidence of this is seen in the visualization in Figure 4, which shows details from the predicted means (soft images). This shows that the DRIU and likelihood baseline overdilate segmentations and the CNF does not. This can be explained from the fact that in the weighted Bernoulli it is cheaper to overdilate than to underdilate. Since the CNF contains no handcrafted loss function, we circumvent this pathology.

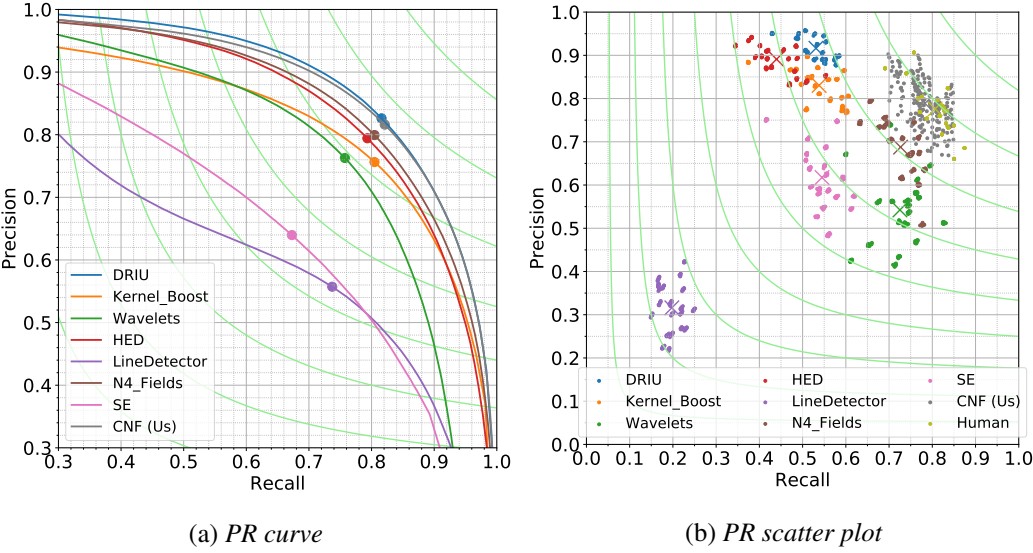

(a) *PR curve*                    (b) *PR scatter plot*

Figure 5: Here we show two visualizations of the same data. LEFT: We show the PR-curves generated from a sweeping threshold on soft images output by each listed method. Maximal F-scores for each curve are shown as circles with the green lines indicating constant F-score. We see that our method beats all traditional methods and is on par with DRIU, which unlike us was pretrained on Imagenet. RIGHT: We show a scatter plot in PR-space of samples drawn from each model. To draw samples from the all factored models, we sample images from a factored Bernoulli with a mean as the soft image. We see that the DRIU and HED models, while having good precision, have poor recall in this regime. This indicates that while the output of their networks produce a good ranking of probabilities, the values of the probabilities are poorly calibrated. For us, we drop in precision slightly, but gain greatly in terms of recall, indicating that our samples are drawn from a better calibrated distribution, overlapping significantly with the *Human* distribution.

## 6   CONCLUSION

In this paper we propose to *learn* likelihoods of conditional distributions using conditional normalizing flows. In this setting, supervised prediction tasks can be framed probabilistically. In addition, we propose a generalization of variational dequantization for binary random variables, which is useful for binary segmentation problems. Experimentally we show competitive performance with competing methods in the domain of super-resolution and binary image segmentation.

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

# A ARCHITECTURES

This section describes architecture and optimization details of the conditional normalizing flow network, low-resolution image feature extractor, and shallow convolutional neural network in the conditional coupling layers.

The conditional coupling layer is shown schematically in Figure 6. This shows that conditioned on an input $\mathbf{x}$, we are able to build a relatively straight-forward invertible mapping between latent representations $\mathbf{z}$ and $\mathbf{y}$, which have been partitioned into vectors of equal dimension.

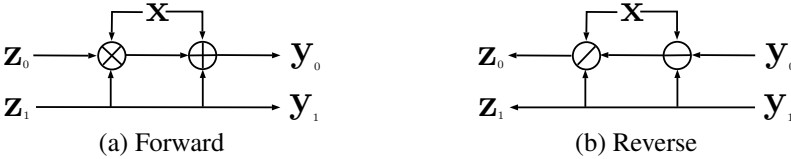

| (a) Forward | (b) Reverse |
|:-:|:-:|

Figure 6: The forward and reverse paths of the conditional coupling layer. In our experiments we concatenate an embedding of the conditioning input $\mathbf{x}$ to the latent $\mathbf{z}_1$, which is fed through another neural network to output the affine transformation parameters applied to $\mathbf{z}_0$. This operation is invertible in $\mathbf{z}$ and $\mathbf{y}$, but not in $\mathbf{x}$.

Details for the CNFs are given in Table 5 and the details of the individual coupling layers in Table 6 and 7. The architecture of the feature extractor is given in Table 8. The architecture has levels and subflows, following (Dinh et al., 2016; Kingma & Dhariwal, 2018). All networks are optimized using Adam (Kingma & Ba, 2015) for 200000 iterations.

Table 5: Configuration of the CNF architecture for the super-resolution task.

| DATASET | MINIBATCH SIZE | LEVELS | SUB-FLOWS | LEARNING RATE |
|:-:|:-:|:-:|:-:|:-:|
| ImageNet32 | 64 | 2 | 8 | 0.0001 |
| ImageNet64 | 64 | 2 | 8 | 0.0001 |
| DRIVE | 2 | 2 | 2 | 0.001 |

Table 6: Architecture details for a single coupling layer in the super resolution task. The variable $c_{\text{out}}$ denotes the number of output channels. The first two convolutional layers are followed by a ReLU activation.

| LAYER | INTERMEDIATE CHANNELS | KERNEL SIZE |
|---|---|---|
| Conv2d | 512 | $3 \times 3$ |
| Conv2d | 512 | $1 \times 1$ |
| Conv2d | $c_{\text{out}}$ | $3 \times 3$ |

Table 7: Architecture details for a single coupling layer in the DRIVE segmentation task. The variable $c_{\text{out}}$ denotes the number of output channels for the . The first two convolutional layers are followed by a ReLU activation.

| LAYER | INTERMEDIATE CHANNELS | KERNEL SIZE |
|---|---|---|
| Conv2d | 32 | $3 \times 3$ |
| InstanceNorm2d | 32 | - |
| ReLU | - | - |
| Conv2d | $c_{\text{out}}$ | $3 \times 3$ |

Table 8: Architecture details for the conditioning network in the super-resolution task. Residual-in-residual denseblocks Wang et al. (2018) are utilized. The channel growth is adjusted so that the CNF and the factorized baseline have an equal number of parameters.

| MODEL TYPE | RRDB BLOCKS | CHANNEL GROWTH | CONTEXT CHANNELS |
|---|---|---|---|
| CNF | 16 | 32 | 128 |
| Factorized LL | 16 | 55 | 128 |

# B    CONDITIONAL IMAGE GENERATION

In this section, larger versions of the ImageNet64 samples are provided, sampled at different temperatures $\tau$.

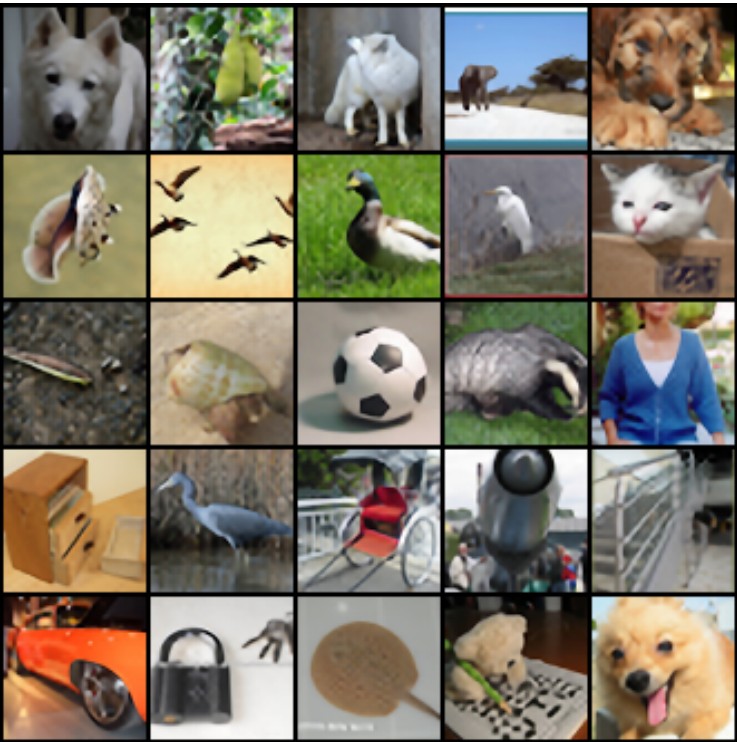

Figure 7: Super resolution results CNF trained on Imagenet64 sampled at temperature $\tau = 0$.

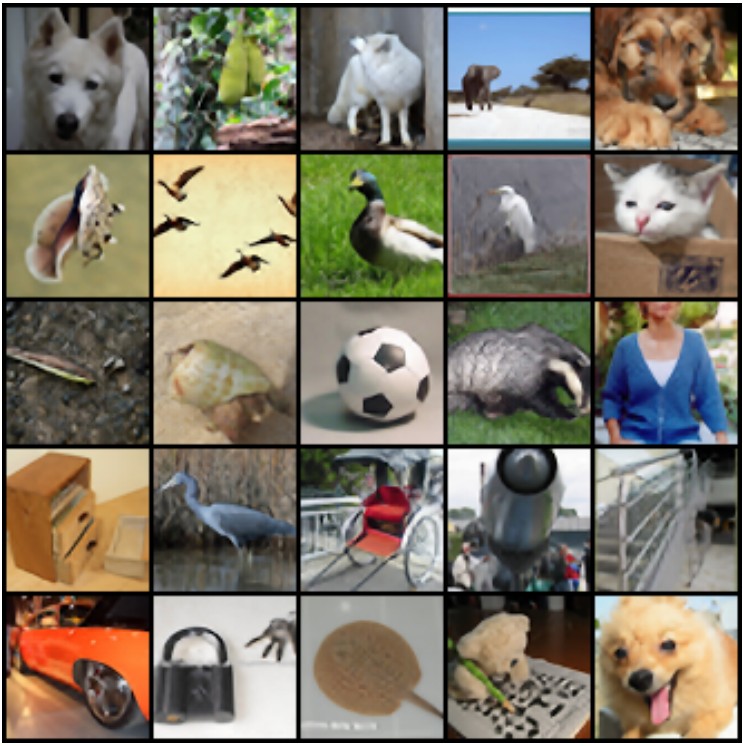

Figure 8: Super resolution results for the CNF trained on Imagenet64 sampled at $\tau = 0.5$

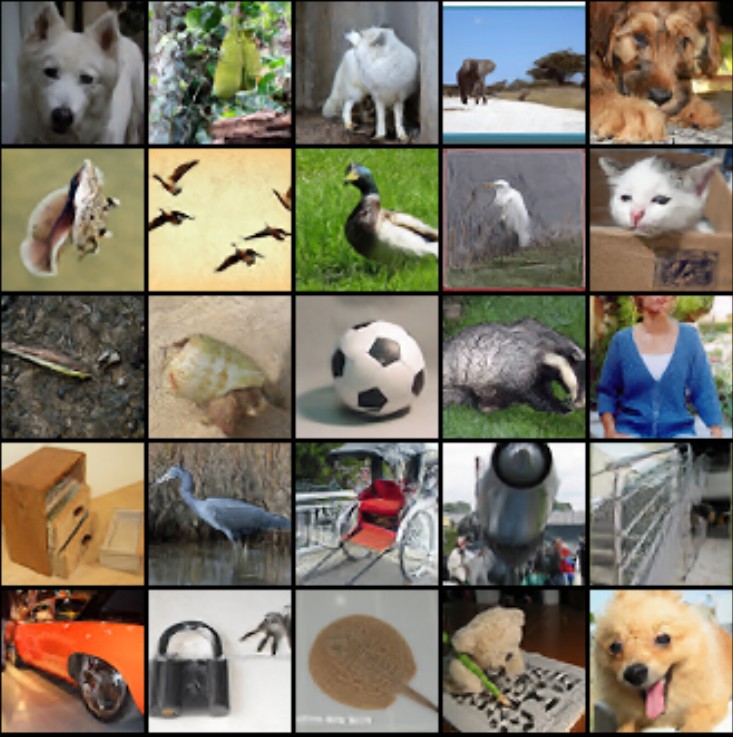

Figure 9: Super resolution results for the CNF trained on Imagenet64 sampled at $\tau = 0.8$

