# OpenReview forum: "Learning Likelihoods with Conditional Normalizing Flows "
_ICLR.cc/2020/Conference — Reject_

### Official Review · AnonReviewer1 · 2019-10-23
**Official Blind Review #1**

**Rating:** 3

**Review:**

The paper proposes the conditional normalizing flow for structured prediction. The idea is to use conditioning variables as additional inputs to the flow parameter forming networks.  The model was demonstrated on image superresolution and vessel segmentation.

I find the contribution of this paper minimal. The idea of conditioning has extensively been used during recent years because it is the most natural thing to do (e.g., [1], [2] and numerous other papers). Their's nothing new about the flows used in this paper. The results in table 2 are not convincing; I see no benefit of using the proposed flow model for image super-resolution instead of the SOTA super-resolution methods. This also applies to other experiments.

[1] van den Oord et al., Conditional Image Generation with PixelCNN Decoders, 2016.
[2] Papamakarios et al., Masked Autoregressive Flow for Density Estimation, 2017.

**Experience Assessment:**

I have read many papers in this area.

**Review Assessment: Checking Correctness Of Derivations And Theory:**

I assessed the sensibility of the derivations and theory.

**Review Assessment: Checking Correctness Of Experiments:**

N/A

**Review Assessment: Thoroughness In Paper Reading:**

N/A

---

> ### Author Response · Authors · 2019-11-14
> **Response to Reviewer 1**
>
> Thank you for your comments.
>
> We would like to disagree on the topic on novelty of our contribution. In particular, reference [1] is not a flow. And indeed as you state, class-conditional flow models are not new in the literature. Where we would like to draw our distinction, however, is that we are in fact not considering class-conditional flow models. Instead our generative flow uses high-dimensional images as the conditioning argument. This warrants the use of a different kind of conditional coupling layer, unlike the ones in the papers that you cite. These papers also happen to be autoregressive, which makes sampling computationally expensive.
>
> Another contribution we would like to highlight, which was also recognized by reviewers 2 and 3, is the link we draw between variational dequantization and existing variational inference methods. This new viewpoint allows us to derive a form of variational dequantization adapted to binary random variables in a consistent probabilistic framework. This innovation is important when it comes to finding a good lower bound on the likelihood. For instance, in the retinal vessel segmentation experiments, the log-likelihood scores for uniform dequantization versus our method is about 0.35 BPD versus 0.025 BPD (2.s.f). In an updated version of the paper, we are going to include this results to stress this improvement.
>
> With respect to Table 2, the specific metrics that are important depend on what task you are willing to solve. In terms of fitting distributions, we outperform our baselines. As stated in our introduction, we want to learn distributions over the data, because they can be easily evaluated in terms of likelihood, they are very interpretable compared to other generative methods such as GANs, there is no mode dropping behavior, and there exists easy tests for overfitting.

---

### Official Review · AnonReviewer3 · 2019-10-25
**Official Blind Review #3**

**Rating:** 6

**Review:**

Summary of the paper:

The paper proposes an extension of Normalizing flows to conditional distributions. The paper is well written overall and easy to follow. Basically the conditional prior z|x = z=f_{\phi}(y,x), where x is the conditioning random variable, and we apply the change of variable formula to get the density of y|x . For example in super resolution y is the high res image and  x is the low res. image.  To sample from the models authors propose to use f^{-1}_{\phi}(z;x).

The conditional modules are natural extensions of invertible blocks used in the literature (coupling layers, split priors, conditional coupling, 1x1 conv), where the conditioning is done on some hidden representations of the conditioning variable x (i.e one or multiple layers of NN).

Authors propose a dequantization for binary random variables (useful for segmentation applications), where they give an implicit model for the dequantizer (obtain a continuous variable from a discrete binary variable).

Author apply the method in two applications super-resolution and vessel segmentation. the method is compared to supervised learning of the corresponds between x and y and to others competitive methods in the literature and shows some advantage.

Minor comments :

- Formatting the bibliography is messed up and needs some cleaning , Figure 5 is also making formatting issues of the paper.
- Figure 1 for sampling it should be f^-1_{\phi } and not f_{\phi}

Review:

- Figure 2 is hard to get any idea of the sample quality would be good also to put the low resolution input to the algorithm . Also did you use a temperature sampling for the baseline ? otherwise the comparison is not fair.

- The Drive database is too small 20 training samples and 20 testing only? can the model be just overfitting?

- In the vessel implementation why do you drop the scaling modules?

- The conditioning for the vessel implementation on x is on two layers , would be great to put all architectures of the models in details , and to show both sampling and training paths

- It would be great to add the details of the skip connection used from the network processing x, and how ensure that the flow remains invertible.

Overall this is a well written paper and a good addition to normalizing flows methods , some discussion of related works on conditional normalizing flows and more baselines with other competitive methods based on GANs for example would be helpful but not necessary.

It would be great to add details of the architectures and on skip connections and how to ensure invertibility for this part in the model .


**Experience Assessment:**

I have read many papers in this area.

**Review Assessment: Checking Correctness Of Derivations And Theory:**

I carefully checked the derivations and theory.

**Review Assessment: Checking Correctness Of Experiments:**

I carefully checked the experiments.

**Review Assessment: Thoroughness In Paper Reading:**

I read the paper thoroughly.

---

> ### Author Response · Authors · 2019-11-14
> **Response to Reviewer 3**
>
> Thank you for your comments
>
> In Figure 2, we will include examples of the low resolution input, for an easier comparison of the results. In this figure in particular, we did not use a temperature for sampling of the baseline, since we are displaying the mode of the distribution. Since the distribution is factorized, sampling would add uncorrelated noise, meaning this comparison is actually skewed in favour of the baseline model.
>
> Concerning the DRIVE database, it indeed has very few images. Since we are training a likelihood based model, it is very easy for us to check for overfitting and early stop accordingly. In practice, we found that the standard data augmentation implied that we do not overfit. Furthermore, since the task is a per-pixel reconstruction task, the effective number of labels is much higher than the number of training images.
>
> In the DRIVE experiments, we dropped the scaling modules since they did not appear to add much benefit to the results.
>
> With regards to the exact architectures we have now placed network architecture tables in the appendix to clear up any confusion. Furthermore, we are adding a diagram of the conditional coupling layers in the appendix, which show the invertibility property clearly.
>
> We have extended our related work on non-flow-based competing methods from the literature. and we have added some extra references on (conditional) normalizing flows as well.
>
> We have already cleaned up the bibliography and any formatting issues, which we had at submission time. Thank you also for the sharp observation regarding the missing ^{-1} in Figure 1.

---

### Official Review · AnonReviewer2 · 2019-10-25
**Official Blind Review #2**

**Rating:** 6

**Review:**

This paper presented the conditional normalizing flows (CNFs) as a new kind of likelihood-based learning objective.  There are two keys in CNFs. One is the parametric mapping function f_{\phi} and the other is the conditional prior. This paper assumed the conditional prior as Gaussian distribution of x. The mapping function is invertible with x as a parameter. The prior parameter and \phi are updated by stochastic gradient descent. The latent variable z is then sampled from conditional prior. The output targe y is obtained with dependency on x and f_{\phi}.

Strength:
1. This study adopted the flow-based model to estimate the conditional flow without using any generative model or adversarial method.
2. This method obtained the advanced results on DRIU dataset without the requirement of pretraining.
3. This paper proposed an useful solution to train continuous CNFs for binary problems.

Weakness:
1. It is required to address how to design the function f_{\phi} which depends on x. In particular, the property invertibility should be clarified.
2. Why the issues of mode collapse or training instability in flow are considerable in the experiments?
3. It will be meaningful to evaluate this method by performing the tasks on text to image or label to image.

**Experience Assessment:**

I have published one or two papers in this area.

**Review Assessment: Checking Correctness Of Derivations And Theory:**

I did not assess the derivations or theory.

**Review Assessment: Checking Correctness Of Experiments:**

I assessed the sensibility of the experiments.

**Review Assessment: Thoroughness In Paper Reading:**

I made a quick assessment of this paper.

---

> ### Author Response · Authors · 2019-11-14
> **Response to Reviewer 2**
>
> Thank you for your comments.
>
> To clarify your concerns, the design is covered in the section 3.1 Conditional modules. In particular the main invertible module is the conditional coupling layer. This takes in a conditioning input x and a latent variable z, which is transformed deterministically into a latent variable y. The transformation y <-> z conditioned on x is invertible. This transformation is similar to the coupling layer of RealNVP, but where every subnetwork in the layer takes and additional x as input. For clarity, we can add a diagram detailing this in the appendix.
>
> With regards to your comments about mode collapse and training instability, it has been noted in the literature that normalizing flows do not suffer so much from mode collapse in the same way that GANs do, for instance. And on the topic of training instability, we did not notice any instabilities in the training of our flow models.
>
> Thank you for your suggestions on follow up experiments. We agree that a text to image scenario would be interesting, since the conditioning argument in this case is structured.

---

### Public Comment · ~Joseph_Marino1 · 2019-10-02
**Great idea, nice set of experiments, and an additional reference**

I'm a strong advocate of moving beyond the limitations of parametric distributions by using normalizing flows, so I'm happy to see the nice set of experiments in this paper. Best of luck! You might consider including the following, somewhat obscure, reference:

Deep Variational Inference Without Pixel-Wise Reconstruction, Agrawal & Dukkipati, 2016, (https://arxiv.org/abs/1611.05209)

They parameterize the conditional likelihood in a variational autoencoder using normalizing flows.

---

> ### Author Response · Authors · 2019-10-14
> **Thank you**
>
> Thank you for your comment and the reference. We will include it in our paper.

---

### Public Comment · ~Lynton_Ardizzone1 · 2019-10-14
**Prior Work**

Hello,

we would like to point out our work, which was published on arxiv on July 4th this year:
"Guided Image Generation with Conditional Invertible Neural Networks" (https://arxiv.org/abs/1907.02392 ),
which is very similar in the approach, and is also applied to an inverse problem in computer vision (colorization).

We feel this should be included in the related work, and differentiated from your own contributions.

Best regards,
Lynton Ardizzone
Visual Learning Lab Heidelberg

---

> ### Author Response · Authors · 2019-10-18
> **Response to prior work**
>
> Hi Lynton,
>
> Thanks for the reference. It looks like a very nice paper. It certainly is relevant and we shall of course include it in our related work.
>
> To answer your question about differences and similarities, here is a brief list:
>
> - Architecture: To some degree our architectures are similar. We do indeed both use conditional affine coupling layers (cACL). Perhaps the largest difference is that you couple two cACLs together; whereas, we use a single cACL followed by a learnable, (nonconditional) 1x1 convolution (you refer to this as a soft channel permutation). Furthermore, we deploy a dequantization network (more below).
>
> - Dequantization: We introduce a new variational dequantization scheme, which builds on the work of Flow++, (Ho et al., 2019). This works for binary data spaces. Furthermore, we make a connection between variational dequantization and variational inference, which allows us to generalize the binning scheme of Flow++.
>
> - Per-pixel loss interpretation: We make explicit the disadvantages of previous per-pixel reconstruction losses, which forms the motivation for why we would wish to use a flow.
>
> - ML versus MAP: We do maximum likelihood, which is well known to be parameterization invariant instead of, say, MAP inference.
>
> We hope this answers your questions. If you have more, do feel free to let us know.
>
> Best,
> The authors

---

> > ### Public Comment · ~Lynton_Ardizzone1 · 2019-10-24
> > **Response to prior work**
> >
> > Hello,
> >
> > thank you for the response!
> >
> > We agree with the differences in architecture and dequantization.
> >
> > Concerning the training:
> > - We also train as a normalizing flow using maximum likelihood training (see our Sec. 3.2).
> > - The main difference we see, is that you use a conditional split prior in addition to the conditional Flow, whereas we only have a conditional flow, with a fixed unconditional prior.
> > - It may be informative performing an ablation, to demonstrate the improvement produced by the more flexible conditional prior.
> >
> > Best,
> > Lynton

---

> > > ### Author Response · Authors · 2019-10-25
> > > **Response to prior work**
> > >
> > > Thanks for acknowledging the differences in the architecture and dequantization.
> > >
> > > We do have one disagreement though, which we would like to flag about the training objective. In Sec 3.2 of your paper, equations 5 and 6 denote the (unnormalized) log-posterior distribution of the weights of your flow given the data. Therefore, it seems from the paper that you are performing maximum a posteriori (MAP) model fitting of your weights.
> > >
> > > Best,
> > > The authors

---

> > > > ### Public Comment · ~Lynton_Ardizzone1 · 2019-10-30
> > > > **Response to prior work**
> > > >
> > > > Hi,
> > > >
> > > > I am sorry in case our notation is confusing, we will change it in a future revision in this case. But I am confident we use the same loss, because we also use the standard maximum likelihood loss used to train normalizing flows. Put simply,
> > > >
> > > > L = 0.5 * z^2 - log(det(J))
> > > >
> > > > With latent vector z and Jacobian J.
> > > > The loss is only applied in z-space, not on the actual images (therefore not MAP).
> > > >
> > > > Your eq. 7 is the same as our eq. 4 (without the conditional split prior), and we directly optimize the negative logarithm of this, in the same way as you.
> > > >
> > > > So I feel you have misunderstood our training procedure. (Perhaps it is also the case, because we use the term 'backwards' and 'fowards' with regards to the flow in the opposite way, calling X -> Z 'forward')

---

> > > > > ### Author Response · Authors · 2019-11-02
> > > > > **MAP vs MLE**
> > > > >
> > > > > Hi Lynton,
> > > > >
> > > > > Thank you for your reply. We agree with you that eq. 4 is the maximum likelihood.
> > > > >
> > > > > However in your paper you say that you minimize the loss as the negative logarithm of:
> > > > > p(theta | x, c) proportional to p(x | c, theta) p(theta). (Eq. 5 & 6 in your paper)
> > > > >
> > > > > The paper refers to this as the "posterior over model parameters". Perhaps you could explain what you think is the difference between MLE and MAP?
> > > > >
> > > > > Best,
> > > > > The authors

---

> > > > > > ### Public Comment · ~Lynton_Ardizzone1 · 2019-11-04
> > > > > > **MAP vs MLE**
> > > > > >
> > > > > > Hello,
> > > > > >
> > > > > > I apologize, I misunderstood what was meant by MAP (I took it to mean the MAP w.r.t. x of p(x|y), as would be learned by a standard feed-forward regression model).
> > > > > > In this case, I agree with the distinction you make.
> > > > > >
> > > > > > So as I understand it, the practical difference between the two training procedures (MAP/MLE) is whether L2 weight regularization is applied to the network weights or not.

---

### Decision · Program_Chairs · 2019-12-19

**Decision:**

Reject

**Comment:**

The authors propose a conditional normalizing flow approach to learning likelihoods. While reviewers appreciated the paper, in its present form it lacked a clear champion, and there were still some remaining concerns about novelty and clarity of presentation. The authors are encouraged to continue with this work and to account for reviewer comments in future revisions. Following up on the author response, a reviewer adds:
"Thanks for your clarification. I still disagree that the conditional flow architecture proposed should be considered as a novel contribution. The reason why I mentioned [1] or [2] was not because they follow the exact setting (coupling based conditional flow model) discussed in this paper. I wanted to highlight that the idea to use conditioning variables as an input to the transforming network (whether it is an autoregressive density function, autoregressive transforming network, or coupling layers) is quite universal (as we all know many of the existing codes implementing flow-based models includes additional keyword arguments 'context' to model conditioning). I'm not sure why the fact that the proposed framework is conditioning on high-dimensional variables makes a contribution. There seems to be no particular challenge in doing that and novel design choices to circumvent that (i.e., we can just use existing architectures with minor modifications).

I agree that the binary dequantization should be considered as a contribution, but as significant as to change my decision to accept. Thanks for the clarification on experiments. Considering this, I raise my rating to weak reject...

Another previous work I forgot to mention in the initial review is "Structured output learning with the conditional generative flow", Lu and Huang 2019, ICML 2019 invertible neural network workshop. This paper discusses the conditional flow based on a similar idea, and attacks high-dimensional structured output prediction. I think this should be cited in the paper."